# Prevention and Modern Strategies for Managing Methicillin-Resistant Staphylococcal Infections in Prosthetic Joint Infections (PJIs)

**DOI:** 10.3390/antibiotics13121151

**Published:** 2024-12-01

**Authors:** Karolina Kraus, Paweł Mikziński, Jarosław Widelski, Emil Paluch

**Affiliations:** 1Faculty of Medicine, Wroclaw Medical University, Wyb. Pasteura 1, 50-376 Wroclaw, Poland; karolina.kraus@student.umw.edu.pl (K.K.); pawel.mikzinski@student.umw.edu.pl (P.M.); 2Department of Pharmacognosy with Medicinal Plants Garden, Lublin Medical University, 20-093 Lublin, Poland; jaroslaw.widelski@umlub.pl; 3Department of Microbiology, Faculty of Medicine, Wroclaw Medical University, Tytusa Chalubinskiego 4, 50-376 Wroclaw, Poland

**Keywords:** *Staphylococcus aureus*, coagulase-negative *staphylococci*, endoprostheses, periprosthetic joint infections (PJIs), antimicrobial treatment, bacteriophage therapy

## Abstract

Periprosthetic joint infections (PJIs) are a dangerous complication of joint replacement surgeries which have become much more common in recent years (mostly hip and knee replacement surgeries). Such a condition can lead to many health issues and often requires reoperation. Staphylococci is a bacterial group most common in terms of the pathogens causing PJIs. *S. aureus* and coagulase-negative staphylococci are found in around two-thirds of PJI cases. Recently, the numbers of staphylococci that cause such infections and that are methicillin-resistant are increasing. This trend leads to difficulties in the treatment and prevention of such infections. That is why MRSA and MRSE groups require extraordinary attention when dealing with PJIs in order to successfully treat them. Controlling carriage, using optimal prosthetic materials, and implementing perioperative antimicrobial prophylaxis are crucial strategies in infection prevention and are as essential as quick diagnosis and effective targeted treatment. The comprehensive professional procedures presented in this review show how to deal with such cases.

## 1. Introduction

Prosthetic joint infections are a destructive complication following hip or knee replacement surgery. With the rise in these medical procedures in recent years, PJIs have become increasingly common [1]. Their clinical presentation is often atypical, with common infection signs, like fever and sepsis, frequently absent. The presence of a prosthesis increases the risk of infection, as it requires a lower microbial load than a native joint. Moreover, a prosthesis as a foreign body itself is more prone to biofilm formation which makes those pathologies even more problematic in treatment. *Staphylococcus aureus* is one of the most frequent culprits in the case of such infections with coagulase-negative staphylococci, especially *S. epidermidis*, also being one of the leading causes [2]. A growing concern is PJIs caused by antibiotic-resistant bacteria. They present a significant therapeutic challenge. Methicillin-resistant *Staphylococcus aureus* (MRSA) and methicillin-resistant *Staphylococcus epidermidis* (MRSE) are among the top concerns in the treatment of PJIs due to antibiotic resistance [3]. Methicillin resistance in the staphylococci group is a mechanism acquired by the presence of the mecA gene which makes such bacteria resistant to beta-lactams. MRSA is a major contributor to hospital-acquired infections (including PJIs), often leading to considerable morbidity, mortality, extended hospital stays, and increased healthcare costs [4]. According to Hays et al., the incidence of MRSA as a cause of PJIs has notably increased in recent years. Its marked increase occurred after 2004, rising from 0 to 5.8% prior to 2004 to 13.6 to 19.2% afterwards [5]. Another study from 2021 by Anagnostakos et al., presented periprosthetic hip and knee joint infections overview in which MRSA and MRSE contributed to 23.7% of the cases [6].

The factors mentioned above show how important the MRSA and MRSE topic is in relation to prosthetic joint infections. In our review, we present and discuss current strategies for the successful prevention and treatment of such infections. Furthermore, we explore modern approaches, including experimental methods such as bacteriophage therapy, which can be particularly useful in cases where standard treatment protocols fail.

## 2. Methicillin-Resistant Staphylococci (MRS)

*Staphylococcus aureus* and CNS—coagulase-negative staphylococci (*Staphylococcus epidermidis*, *S. haemolyticus*, *S. saprophyticus*)—can develop multiple drug-resistant mechanisms, one of them being methicillin-resistance. These mechanisms make such bacteria unsusceptible to several common antimicrobial agents and result in infections caused by them being much more challenging [7]. Methicillin resistance in *Staphylococcus* strains is due to the staphylococcal cassette chromosome (SCC*mec*), which includes the *mec* gene for resistance and the ccr complex for integrating and removing the cassette from the genome [8]. When it comes to *mec* genes, we can identify *mecA*, *mecB*, *mecC*, and *mecD* [9]. The *mecA* and *mecC* genes, which encode variants of PBP2a, are particularly important. Their presence enables the detection of MRSA using rapid PCR tests designed to target these genes [10]. The mechanism of methicillin resistance is based on the modification of a protein crucial for beta-lactams antimicrobial activity. Methicillin-resistant staphylococcal isolates express an additional penicillin-binding protein 2 (PBP2) in place of regular PBP. Normally, this protein (PBP) facilitates the cross-linking or transpeptidation processes that connect peptidoglycan chains within the bacterial cell wall. β-lactams serve as inhibitors towards this protein which leads to the weakening of bacterial walls (the reason for β-lactams antimicrobial activity). PBP2 does not react in this way which makes such bacterial isolates resistant to β-lactams (penicillins, most cephalosporins, and carbapenems) [11]. Although there are antimicrobial agents to combat methicillin-resistant isolates, such as vancomycin, daptomycin, linezolid, clindamycin, tedizolid, trimethoprim-sulfamethoxazole, ceftaroline, ceftobiprole, delafloxacin and other less common agents, the mortality of those infections is high [12]. While methicillin-resistant *Staphylococcus aureus* (MRSA) poses a significant challenge in various fields of medicine, it is important to underline the fact that coagulase-negative staphylococci, common components of the skin flora, can also develop methicillin resistance, making them highly dangerous opportunistic pathogens [13].

## 3. Prosthetic Joint Infections (PJIs), Biofilms, and MRSA

Prosthetic joint infections (PJIs) are a challenging complication affecting orthopedic patients after joint replacement surgeries. In recent years, their worldwide occurrence has risen significantly due to the increasing numbers of such procedures [2].

PJI symptoms vary depending on the timing. Acute infections appear within weeks to months post-surgery with pain, redness, and swelling, while hematogenous infections occur years later due to bacteremia. Chronic PJIs usually present within two years, primarily as pain, and can be mistaken for noninfectious issues like wear or loosening [1].

Bacterial biofilm formation is a key factor in the severity of these infections. Endoprostheses, as foreign bodies, are prone to biofilm formation during infection. Biofilm is a multi-species microbial structure that offers protection against harsh conditions like drying and starvation. Its clinical resistance to antibiotics is heightened by the presence of cells with low metabolic activity, known as persister cells [14]. These factors make PJIs especially difficult to treat without complex surgical interventions.

To combat biofilm-forming bacteria, especially in chronic infections, prolonged antibiotic therapy is often required. In many cases, treatment involves the removal of the endoprosthesis followed by reoperation [15]. Another significant challenge is the rise in antibiotic-resistant bacteria, with MRSA being an increasingly common culprit [5].

## 4. Modern Prosthetic Material Modifications Targeted at Preventing MRSA

When fast-growing biofilm-forming bacteria, such as MRSA and MRSE, attach to the implant material, the process of biofilm formation is initiated. The period from the initial bacterial adhesion to the point where the attachment becomes irreversible and the early biofilm stage develops is considered a critical “therapeutic window” for eliminating periprosthetic joint infections (PJIs) before the biofilm is fully established. At this stage, bacteria remain susceptible to conventional antibiotics and immune system responses [16]. There are two main strategies for improving the materials used in prosthetics, preventing biofilm adhesion through passive surface modification/finishing (PSM) and eradicating microorganisms via active surface modification/finishing (ASM). A third approach, which does not directly involve modifying the implant, includes local carriers or coatings (LCCs), applied during the surgical procedure, either immediately before or simultaneously with the implant, and around it. All commonly used orthopedic materials, such as cobalt-chromium, titanium, polyethylene, polymethyl methacrylate (PMMA), and ceramics, are susceptible to colonization by biofilm-forming bacteria [17,18].

Van Dun et al. tested various surfaces, including stainless steel, titanium, and different types of polymers, to observe how these materials impact the formation and growth of MRSA biofilms. The results indicate that the type of material significantly affects biofilm development. Surfaces that are rougher or have certain chemical properties tend to promote stronger biofilm formation. For example, polymers with hydrophobic characteristics facilitated more substantial MRSA biofilm growth compared to metals like stainless steel and titanium, which are generally less conducive to biofilm formation. The study found that the susceptibility of MRSA biofilms to antibiotics remained consistent across the different implant materials tested, suggesting that the effectiveness of antibiotics was not influenced by the surface roughness of the biofilms. Despite variations in biofilm formation based on the material type, these differences did not impact the ability of antibiotics to penetrate and act against the biofilms. This implies that while the choice of material may affect biofilm development, it does not alter the overall antibiotic efficacy in treating MRSA biofilms on these surfaces [19].

It is worth noting that a current trend is the search for additives to commonly used materials that aim to enhance the effectiveness of preventing *Staphylococcus* spp. infections especially MRSA.

Laminating bioactive implants with transition metal complexes can significantly reduce the virulence of MRSA. These metal complexes, such as those containing silver, copper, or zinc, are known for their antimicrobial properties. When incorporated into the implant surface, they interact with bacterial cells in several ways. Transition metals can generate reactive oxygen species or directly interact with the bacterial cell membrane, leading to its destabilization. The metal complexes interfere with the bacteria’s ability to form biofilms, which are protective layers that MRSA and other bacteria produce to shield themselves from the immune system and antibiotics. Transition metals can also interfere with critical metabolic functions within bacterial cells by binding to enzymes and proteins that are essential for their survival. These metal complexes can release metal ions gradually, creating a sustained antibacterial environment around the implant. The continuous release of metal ions ensures long-term protection against bacterial colonization and infection. Together, these mechanisms make the transition metal complex-coated implants highly effective at reducing MRSA virulence [20]. These implants not only prevent bacteria from attaching and growing but also actively weaken or eliminate any bacteria that come into contact with their surface [21].

Palladium (II) thiazolinyl picolinamide complex (Pd(II)-E) has proven to be an innovative, effective antimicrobial agent. The findings indicate that Pd(II)-E excels in targeting MRSA by effectively disrupting biofilm formation and countering the various virulent mechanisms including slime synthesis, phenol soluble modulin (PSM)-mediated spreading, exopolysaccharide production, and staphyloxanthin synthesis. The results underscore the potential of Pd(II)-E as a valuable tool in managing and mitigating MRSA-related infections [22].

As is commonly known, biomaterials containing copper have been shown to exhibit antimicrobial, angiogenic, osteogenic, and tissue-healing properties. The in vitro studies on antibacterial and immunological properties reveal that Cu-incorporated sulfonated poly(ether ketone) (SPEEK) exhibits an effective bactericidal action against MRSA through both “trap killing” and “contact killing” mechanisms. The role of SPEEK is to increase the hydrophilicity, enhance ion exchange properties, improve biocompatibility, and exhibit antibacterial properties when combined with copper. Additionally, macrophages cultured on a Cu-incorporated SPEEK show activation and polarization towards a pro-inflammatory phenotype, along with enhanced phagocytic activity against MRSA. Further evidence from in vivo infection models associated with implants confirms the superior antibacterial efficacy of Cu-incorporated SPEEK. These findings highlight the multifaceted antibacterial effects of Cu-incorporated SPEEK [23].

Lysostaphin is an enzyme that degrades the cell walls of the Staphylococcus bacteria, rendering it a potent agent against infections caused by methicillin-resistant *Staphylococcus aureus* (MRSA). In a study conducted by Jaekel et al., the efficacy of lysostaphin-coated titanium plates was evaluated in a minipig model of osteitis induced by MRSA. The findings demonstrated that the lysostaphin coating significantly reduced the bacterial load on the implant surfaces, leading to a marked reduction in infection risk compared to uncoated implants. Nevertheless, the stability of the PDLLA-lysostaphin coating has not yet been assessed and animal experiments cannot entirely replicate the intricate immunological and biomechanical conditions of the human body [24].

Nitric oxide (II) (NO) has long been considered an excellent method for combating MRSA biofilm infections [25]. NO effectively combats prosthetic joint infections (PJIs) by directly killing bacteria, disrupting biofilms, and enhancing immune responses, particularly through the activation of macrophages. Additionally, NO supports implant integration by promoting bone growth and reducing infection risk. A novel and innovative concept is the use of light-controlled NO release. In their study, Li et al. treated bone implant biofilm infections using a hydrophilic and viscous hydrogel of polyvinyl alcohol modified with chitosan and polydopamine, and an NO release donor was formed on red phosphorous nanofilm deposited on a titanium implant (Ti-RP/PCP/RSNO) that released controlled amounts of NO upon 808 nm irradiation in vitro, combined with additional immunotherapy in vivo. The cross-linked PDA and CS enhanced the mechanical and physical antimicrobial properties of Ti-RP/PCP/RSNO even without light exposure. After incubation and irradiation, more than 93.1% of the MRSA biofilm was eradicated, while 89.6% of the biofilm was inhibited after 2 days of irradiation and incubation. The underlying mechanism involved a synergy of ·ONOO−, hyperthermia, ·O2−, and immunological reactions triggered by NIR light. The in vivo evaluations of anti-biofilm activity and bone integration demonstrated that the MRSA biofilm was effectively removed through a combination of photothermal bacteria-killing, NO therapy, and immunological treatment, simultaneously achieving excellent bone formation [26].

## 5. *Staphylococcus aureus* Decolonization

The issue of carrying *Staphylococcus* spp. bacteria is frequently discussed in many specialized centers around the world. The nasal cavity is a potential source of bacterial contamination leading to surgical site infections (SSI). About 20% to 30% of the general orthopedic population are carriers of *S. aureus* [27]. A meta-analysis from 2020 demonstrated that routinely conducting nasal screenings and performing selective decolonization can greatly reduce the risk of surgical site infections (SSIs) and PJIs after total joint arthroplasty (TJA) [28]. The typical method for *S. aureus* decolonization involves applying mupirocin topically to both nostrils twice daily, often along with daily chlorhexidine showers or skin wipes, for 5 days leading up to the total joint arthroplasty (TJA) [29]. Standard decolonization is cost-effective which is considered a major advantage in many national guidelines [30]. Since 2016, the WHO has recommended *S. aureus* nasal screening and decolonization for orthopedic surgeries [31]. However, a French study from a hospital which implemented the strategy recommended by the WHO showed that in their cohort of patients who had planned orthopedic surgery, the implementation of the *S. aureus* screening and decolonization strategy did not result in a significant reduction in rates of monomicrobial *S. aureus* surgical site infections (0.3% or 7 out of 2305 vs. 0.5% or 9 out of 1926) compared to the period before the strategy was put into place [32]. Economedes et al., in their pilot study, showed through their collected data that out of the 58 patients decolonized before surgery, 39 (67%) tested negative on retesting, while 19 (33%) remained positive for *S. aureus* colonization. Among those 19 patients, 17 (89%) were still colonized by bacteria with the same antibiotic sensitivity profile [33]. In a study by Sousa et al., the authors contend that the existing evidence is insufficient to conclusively endorse *S. aureus* screening and decolonization as a reliable method for preventing infections in total joint arthroplasty. They also point out the difficulty in determining whether an infection that occurred was caused by nasal carriage or if it originated from another source [34].

Additionally, to reduce the risk of MRSA PJI infection resulting from patient colonization, it is important to consider the potential MRSA colonization of the throat and oral cavity. The colonization of the throat also indicates that the patient’s MRSA carriage may be long-term. Studies have suggested that the colonization of these areas can lead to re-colonization of the nasopharynx even after successful decolonization efforts [35]. For this reason, some studies recommend that standard screening protocols include at least two swabs, a nasal swab and an additional throat swab [36]. A study by Lindgren et al. found that the best results for eradicating MRSA from the throat were achieved with a treatment regimen consisting of oral rifampicin combined with either clindamycin or trimethoprim/sulfamethoxazole (SXT) for 7 days, along with nasal mupirocin [37].

Another important aspect worthy of attention is the nasal carriage of *Staphylococcus* spp. by health care workers. Barbos et al. in their study showed that approximately 40% of the screened orthopedic surgical team members were carriers of *S. aureus*. The study compared SSI rates in two groups of patients—those operated on before and after the decolonization of the surgical team. Before decolonization, there were 6 SSI cases per 1000 operations. After decolonization, no SSIs were reported in the 300 cases studied [38]. Another study, carried out in the university hospital among orthopedic residents by Castro-Martinez et al., also confirmed that the colonization of surgeons is a serious problem and introduced the proper strategy. Eradication was achieved in 61.53% of participants who adhered to the decolonization protocol. Among orthopedic surgeons, decolonization strategies using mupirocin, with or without the addition of chlorhexidine, proved effective in eliminating *S. aureus* colonization [39]. Kawamura et al. emphasized that combining nasal decolonization with stringent hand hygiene practices was essential for achieving an optimal reduction in SSIs. Hand hygiene alone was less effective without concurrent nasal decolonization. The intervention at a Japanese tertiary-care hospital led to a reduction in the incidence of SSIs from 0.67% to 0.19%, underscoring the effectiveness of the combined decolonization and hand hygiene strategy in controlling MRSA infections in a surgical setting [40].

## 6. Antibiotic Therapy

### 6.1. MRSA- and MRSE-Targeted Antibiotics

*S. epidermidis* strains exhibit a considerably higher level of methicillin resistance than *S. aureus* strains. The methicillin-resistant *Staphylococcus* spp. prosthetic joint infections (PJIs) typically necessitate intravenous therapy, which commonly includes vancomycin or daptomycin combined with another antibiotic that aligns with the antibiogram, such as rifampin, minocycline, or linezolid [41,42]. It is worth highlighting the reports on the efficacy of dalbavancin in treating periprosthetic joint infections (PJIs) caused by MRSA. Due to its long half-life, dalbavancin simplifies treatment regimens by reducing the frequency of dosing compared to traditional approaches. Combined with its excellent penetration into bone and surrounding tissues and its antibiofilm activity, it appears to be an ideal alternative to conventional therapy [43,44]. Clinical studies have confirmed its effectiveness, supporting its use as a streamlined and potent option for managing complex infections such as PJIs [45,46]. The selection of antibiotics should always be based on the results of the antibiogram, thereby minimizing the unnecessary burden of ineffective antibiotics on the organism and reducing the risk of developing multidrug-resistant strains [47]. However, all empirical treatment strategies proposed for the prevention and management of periprosthetic joint infections (PJIs) should incorporate a systemic antibiotic effective against methicillin-resistant *Staphylococcus* spp. (commonly daptomycin or vancomycin). Indeed, this approach is recommended and supported by numerous studies conducted worldwide [48]. Delafloxacin, a recently FDA-approved oral/intravenous fluoroquinolone, can also be used to treat prosthetic joint infections caused by methicillin-resistant Staphylococci. A study by O’Riordan et al. showed that delafloxacin was comparable to the IV vancomycin + aztreonam combination for treating MRSA skin infections, with good tolerability [49]. Delafloxacin has also been successfully used to treat a PJI caused by MDR *Staphylococcus epidermidis* [50].

### 6.2. Antibiofilm Activity

The ability to form biofilms is a crucial factor in MRSA and MRSE virulence. In addition, Chen et al. found that MRSA isolates from periprosthetic joint infections (PJIs) had a higher biofilm-forming ability compared to MRSA isolates from skin and soft tissue infections (SSTIs). This suggests that the MRSA strains isolated from PJIs may be more effective at colonizing the host and evading its immune response than those isolated from SSTIs [51]. During treatment, the choice of antibiotic should be guided not only by the antibiogram but also by the antibiotic’s ability to penetrate biofilm, as well as its penetration into bones, joints, and surrounding tissues [52]. The commonly used groups of antibiotics are fluoroquinolones, glycopeptides, and aminoglycosides, as well as lipopeptides like daptomycin, rifamycins, and oxazolidinones (linezolid). The efficacy of antibiotics in biofilms can be evaluated using metrics such as the minimal biofilm inhibitory concentration (MBIC) and the minimal biofilm eradication concentration (MBEC) [53]. The results of the study of Mirzeai et al. demonstrate that sub-MIC concentrations of cloxacillin, cefazolin, and clindamycin significantly induce biofilm formation in MRSE strains. Their findings show that the extent of biofilm formation induced by these antibiotics is specific to each antibiotic and is inversely proportional to the amount of biofilm produced by the test strain. This underscores the importance of maintaining appropriate antibiotic concentrations during therapy [54]. The penetration of antibiotics potentially used in PJIs caused by MRSA and MRSE through the *Staphylococcus* spp. biofilm extracellular matrix and penetration into the bone and joint has been summarized in Table 1 [55,56,57].

### 6.3. The Role of Antibiotic Therapy in Surgical Strategies

The choice of surgical strategy depends on various factors, including the duration of infection, the stability of the prosthesis, the condition of surrounding tissues, and the patient’s overall health. The main surgical options include debridement, antibiotics, implant retention (DAIR), one-stage exchange arthroplasty, and two-stage exchange arthroplasty. Surgical treatment should run parallel to antibiotic therapy to achieve the best results. Figure 1 presents the standard treatment protocol for PJIs concerning infections caused by methicillin-resistant *Staphylococcus* spp.

**DAIR** is generally considered for early post-operative PJIs (within 4 weeks of initial surgery) or acute hematogenous PJIs where the infection is detected within 2–4 weeks of symptom onset. The prosthesis should be stable, and there should be no signs of implant loosening. The surrounding soft tissue should be in good condition, without significant abscess formation or extensive necrosis. The effectiveness of DAIR in cases of MRSA and MRSE is very limited, not making it a recommended solution for such infections [58,59]. Following surgery, an extended course of targeted intravenous antibiotics is administered, typically for 4–6 weeks. If rifampin can be used, it is often combined with another antibiotic due to its biofilm-penetrating properties [60]. Sheper et al. in their study demonstrated that the outcomes of treatment for staphylococcal periprosthetic joint infections (PJIs) were similar when comparing the effectiveness of DAIR therapy using rifampin for MRSA and MSSA PJIs with success rates of 58% and 60%, respectively. Their meta-analysis also indicated that rifampicin may have a limited impact on preventing treatment failures [42]. Lora-Tamayo et al. also confirmed that DAIR might have similar effectiveness in MRSA PJIs compared to MSSA PJIs. However, a direct comparison of the results is challenging because they used different antibiotic treatment regimens. The specific combination of quinolones plus rifampin is considered the treatment of choice for an MSSA-PJI. As a result, rifampin combinations for an MRSA-PJI did not prevent failure as effectively as rifampin–fluoroquinolone combinations did in MSSA cases, with failures occurring until the antibiotics were discontinued. Additionally, the development of resistance to rifampin was less common when rifampin was combined with quinolones compared to other antimicrobials. They also highlighted the significant relationship between the timing of the procedure and its effectiveness, noting that their study included a group of hematogenous cases and prostheses placed more than 90 days before the onset of symptoms, which contributed to a lower success rate. The risk of failure is higher in chronic infections, with biofilm formation posing a significant challenge [61]. Tatarelli et al. demonstrated a 94% success rate in their study for treating PJIs with a short course of antibiotic therapy as part of DAIR for an early (lasting up to one month) PJI. They also pointed out that the success of DAIR diminishes over time after surgery because of biofilm formation and the spread of the infection to the surrounding tissues [62].

**One-stage exchange** arthroplasty can be considered for some PJIs, but it is less commonly performed for MRSA due to the high risk of reinfection. It is typically reserved for patients with good soft tissue conditions, a healthy immune system, and a pathogen that is known and susceptible to antibiotics [63].

**Two-stage exchange** is considered the “gold standard” for managing chronic PJIs, especially in cases involving MRSA. It is most appropriate for patients with well-established infections, significant bone loss, or poor soft tissue condition. It is indicated when DAIR has failed, in cases of recurrent infection, or when biofilm-forming organisms like MRSA are involved. The first stage usually involves the removal of the infected prosthesis, debridement, placement of an antibiotic-loaded spacer, and administration of intravenous antibiotics. A commonly recommended approach involves administering antibiotics for a period of 2 to 6 weeks, guided by culture results and tailored to MRSA sensitivities (e.g., vancomycin, daptomycin). The antibiotic regimen may also include rifampin if the strain is susceptible. This typically includes intravenous (IV) antibiotic treatment for 2 to 6 weeks, followed by an additional 2 to 6 weeks of oral antibiotics. After this, antibiotics are usually discontinued for 2 to 8 weeks before reimplantation. This regimen has been widely used and has shown effective infection control outcomes [64]. The second stage consists of reimplanting a new prosthesis and then continuing with postoperative antibiotic therapy. Two-stage exchange is associated with the highest success rates for MRSA PJIs, typically between 70 and 90%, depending on factors like patient comorbidities, surgical technique, and adherence to antibiotic protocols. Hernandez et al. reported greater success with a two-stage revision arthroplasty (100% at 5 years) versus the DAIR procedure (61% at 5 years) [65]. Leung et al. reported a 79% infection control rate for MRSA/MRSE-related hip prosthetic joint infections (PJIs) treated with a two-stage revision procedure [66]. Santoso et al., in their retrospective study, predominantly focused on cases involving MRSA and MRSE, evaluating the effectiveness of two-stage hip joint revision, and achieved an overall infection control rate of 71.4% at the final follow-up. What prompted them to emphasize the need for future modifications in treatment was the observed challenges and outcomes in managing MRSA and MRSE infections [67].

## 7. Antibiotic-Loaded Bone Cements (ALBCs)

To minimize the incidence of PJIs following primary or revision total joint arthroplasty, antibiotic-loaded bone cements (ALBCs) can be used for prosthesis fixation. ALBCs can be prepared using one of the following two methods: manual mixing at the time of implantation, or pre-mixed formulations provided by manufacturers, ready for immediate use. It should also be noted that an excessive amount of added antibiotic can weaken the structure of the cement. ALBCs can be used in both primary arthroplasty and revision surgery due to complications. In primary procedures, the antibiotic included in the cement typically covers a broad spectrum of bacteria, including methicillin-resistant Staphylococcus strains. However, not all authors agree on the necessity of using this technique in primary surgeries, as it may contribute to the development of multidrug-resistant bacterial strains in the operated joint, including MRSA and MRSE [68]. Among the antibiotics used in ABCSs for targeted treatment in revision surgery when MRSA and MRSE are detected, vancomycin, daptomycin, gentamicin, and linezolid are commonly used, either individually or in combination [69].

Snir et al. evaluated the incorporation of 1 g of linezolid, vancomycin, and gentamicin into 40 g of PMMA cement, each used individually. Their findings revealed that linezolid had a significantly longer growth inhibition time (GIT) compared to vancomycin and gentamicin against MRSA and *S. epidermidis*. Consequently, they concluded that linezolid is more effective as a local monotherapy than the other two antibiotics tested [70].

Hsu et al. tested the effect of daptomycin at various concentrations to evaluate its impact on both effectiveness and the mechanical strength of cement. They used 0.5, 1, and 2 g of daptomycin per 40 g of PMMA. Mechanical tests showed that all concentrations did not weaken the mechanical properties of the material. They assessed the percentage of daptomycin eluted over 2 weeks, which was 9.59 ± 0.85%, 15.25 ± 0.69%, and 20.64 ± 20.33% for the respective concentrations. Nonetheless, these results did not impact the effectiveness, and the lower doses were adequate in terms of bioactivity and resistance development [71].

Daptomycin was also investigated by Kuo et al. in a small study involving 22 patients with PJIs focusing on methicillin-resistant Staphylococcus—10 MRSA, 8 MRSE, and 4 MRCoNS. If a methicillin-resistant microorganism was identified at the time of resection arthroplasty, 4 g of daptomycin was incorporated into 40 g of bone cement to achieve therapeutic concentrations in the joint fluid. During the second stage, the antibiotic-loaded bone cement spacer or beads were removed, and intraoperative tissue samples were cultured. Reimplantation was performed after a 2-week antibiotic interval, with normalization of ESR and CRP levels. Following prophylactic intravenous administration of vancomycin, the prosthesis was reimplanted using 1 g of daptomycin in 40 g of bone cement, preserving the mechanical integrity of the cement for knee or hip prosthesis fixation. This regimen resulted in a 100% infection control rate, with a mean follow-up of 2.8 years [72].

Cara et al. evaluated, in their in vitro study, the effectiveness of antibiotic combinations in ALBCs versus the effectiveness of ALBCs loaded with a single antibiotic. They compared commercial ready-to-use small dose cements containing gentamicin alone, gentamicin plus vancomycin, and gentamicin plus clindamycin against plain cement with no antibiotics. They measured the prophylactic anti-biofilm effects of ALBCs against MRSA and MRSE strains. For MRSA, the combination of gentamicin plus vancomycin and gentamicin plus clindamycin demonstrated greater efficacy over time compared to gentamicin alone. For MRSE, the results were similar, but gentamicin alone achieved effectiveness comparable to the other two combinations [73].

However, in the case report by Schmolders et al., the authors described a 52-year-old male patient who had undergone total knee arthroplasty and subsequently developed complications, which led to the explantation of the prosthesis and debridement with a spacer containing vancomycin and gentamicin. The patient presented with MRSE in the operated area despite the use of a gentamicin plus vancomycin-loaded cement spacer. The authors expressed concerns that the use of gentamicin-containing PMMA during the primary surgery could contribute to the development of resistant strains [74].

Furthermore, a recent meta-analysis by Kato et al., which aimed to develop Japanese guidelines for MRSA treatment, found that there was no significant preventative effect of ALBCs for MRSA-related surgical site infections (SSIs). Although the use of ALBCs significantly reduced the incidence of deep SSIs in overall patients, the evidence level was very low [75].

There are still insufficient data clearly indicating when it is beneficial to use ALBCs during primary arthroplasty. It is certainly an effective method for managing complications; however, it requires the development of specific, standardized protocols to improve its effectiveness especially in infectious drug-resistant strains.

## 8. Antibiotic-Loaded Hydrogel

There are some innovative solutions that serve as promising alternatives to bone cement for the localized delivery of antibiotics, aimed at preventing and treating periprosthetic joint infections (PJIs). Among the most promising technologies are biodegradable materials, which can deliver the entire drug dose and avoid the need for subsequent surgeries to remove the foreign body. The initial concentration of antibiotics released from the bone cement is relatively high for the first few weeks but then decreases to subinhibitory levels over time, increasing the risk of resistance development. In contrast, the self-initiated degradation of hydrogel materials has the potential to mitigate this issue [76]. There are reports of the effectiveness of using antibiotic-loaded hydrogels in studies conducted on animals so far.

The in vivo study conducted by Ter Boo et al. on a rabbit model aimed to determine the feasibility of using a gentamicin-loaded thermo-responsive hydrogel against gentamicin-sensitive *Staphylococcus aureus*, specifically in the context of bone healing. The gentamicin-loaded poly(N-isopropylacrylamide) grafted hyaluronic acid (HApN) hydrogel effectively prevented infection and not affect fracture healing. Additionally, this hydrogel did not exhibit any adverse systemic effects and did not trigger any systemic reactions [77].

An in vivo study conducted by Boot et al. in 2021 used a sheep model to investigate the effectiveness of a hyaluronic acid hydrogel loaded with gentamicin and vancomycin in eradicating chronic methicillin-resistant *Staphylococcus aureus* orthopedic infections. All samples from the sheep treated with antibiotic-loaded hydrogel were culture-negative at euthanasia. They emphasized that using a hydrogel for antibiotic delivery offers numerous benefits for treating orthopedic device-related infections (ODRIs). It achieves high local antibiotic concentrations at the infection site while maintaining safe systemic levels. The material is easy to handle, degrades over time, and allows for multiple administrations. In their in vivo study, the hydrogel resulted in the complete eradication of the infection, and the results were significantly better compared to those from an ALBC spacer loaded with gentamicin and vancomycin [78].

Boot et al. conducted a preclinical, in vivo animal study demonstrating the significant effectiveness of using a hydrogel for the local delivery of gentamicin compared to systemic treatment for preventing and treating MSSA orthopedic device-related infections (ODRIs), without notable toxic effects. The hydrogel’s versatility facilitates its easy application and achieves high local concentrations of the antibiotic, making it suitable for both prophylactic and therapeutic purposes in contrast to the control group receiving only systemic treatment [79]. These promising characteristics position the hydrogel as a strong candidate for future studies.

The potential application of hydrogels in clinical trials also relevant for the eradication of methicillin-resistant *Staphylococcus* spp. appears promising and in preclinical, in vivo studies that investigate their ability to incorporate innovative antimicrobials. This includes the integration of bacteriophages [80,81] and phage-derived enzymes, such as peptidoglycan hydrolases [82]. Such developments could greatly enhance the therapeutic effectiveness of hydrogels in managing infections, especially MRSA and MRSE. All advantages of the possible use of a loaded hydrogel with example loaded components against methicillin-resistant *Staphylococcus* spp. are summarized in Figure 2.

## 9. Continuous Local Antibiotics Perfusion Therapy (CLAP)

Recently, a new treatment approach involving continuous local antibiotic perfusion (CLAP) for managing bone and soft tissue infections has been introduced for use. CLAP therapy is regarded as a novel system that enables prolonged maintenance of high antibiotic levels directly at the site of infection. During CLAP, an installation is used that facilitates the continuous flow of antibiotics at the optimal dose through the structures surrounding the joint, as presented in Figure 3. It is increasingly used in the treatment of fracture complications and demonstrated encouraging therapeutic results during efforts to implement it in the treatment of periprosthetic joint infections (PJIs) [83,84].

Gentamicin is one of the antibiotics known for its efficacy against MRSA/MRSE and is considered suitable for localized treatment. The adverse effects commonly associated with aminoglycosides, such as renal impairment and ototoxicity, may potentially be alleviated through the use of CLAP therapy [83]. Ando et al. investigated the use of high-concentration and prolonged gentamicin treatment against MRSA isolates that possess aminoglycoside-modifying enzyme genes. They exposed MRSA biofilms to gentamicin at a concentration of 1200 µg/mL, the same concentration used in current CLAP treatment, and monitored bacterial survival in the biofilm over time. In this experiment, minimum biofilm eradication concentration (MBEC) values were defined as the minimum biofilm eradication concentration for overnight (approximately 16 h) antimicrobial exposure. The results indicated that despite MRSA clinical isolates having an MBEC of >2048 µg/mL of gentamicin, exposure to 1200 µg/mL for 48 h killed about 60% of the strains, and after 72 h, around 80% were killed. Additionally, exposure to gentamicin at 128 µg/mL also eradicated MRSA in the biofilm over time. This suggests that even if the gentamicin concentration at the infection site is lower than the MBEC, the biofilm can still be eradicated with prolonged exposure. It can be inferred that CLAP therapy is effective in controlling MRSA biofilms, as it delivers high concentrations of gentamicin locally and maintains exposure over an extended period [85].

So far, all studies on the effectiveness of CLAP in treating PJIs are based on small sample sizes. Miyake et al. characterized the treatment approach for eight patients with various types of PJIs. They employed DAIR in six patients and a two-stage revision in two patients. The patients received CLAP for an average duration of 8.5 days, along with standard intravenous antibiotic therapy for an average of 22.4 days, followed by oral antibiotics. None of the patients required debridement, and there were no observed recurrences of infection or adverse effects related to the gentamicin or the use of CLAP [86]. Zenke et al. analyzed six cases of CLAP usage in periprosthetic joint infections (PJIs), incorporating both intrajoint antibiotic perfusion (iJAP) and intramedullary antibiotic perfusion (iMAP). They highlighted that CLAP could revolutionize the treatment of bone and joint infections by allowing precise control over the local concentration, distribution area, and duration—factors that conventional local administration cannot manage. The topical concentration of gentamicin sulfate was consistent across all cases, at 1200 µg/mL for each route. iJAP was performed in all cases, with an average duration of 19.3 days (ranging from 15 to 28 days). iMAP was conducted in six patients, with an average duration of 10.0 days (ranging from 5 to 16 days). They emphasized the importance of inducing negative pressure with iSAP or iJAP tubes, as this has been confirmed to be crucial. Two of the six patients experienced a recurrence of infection and required repeat CLAP treatment, but all were able to retain their implants. One patient developed renal dysfunction [87]. Kosugi et al. emphasized the need for monitoring blood antibiotic concentrations during CLAP therapy (gentamicin with a concentration of 2.0 μg/mL was used) to detect potentially toxic levels and prevent adverse effects. In their study, they adjusted the doses used in CLAP based on blood concentration results from samples taken three days after surgery. If the blood concentration exceeded 1 μg/mL, the antibiotic concentration was reduced to 1200 μg/mL to prevent the adverse effects of the therapy [88]. CLAP therapy holds significant promise, particularly for certain patient groups, including those with chronic infections, those who have not responded to previous treatments, and those who are unable to tolerate high doses of systemic antibiotics, e.g., with renal dysfunction [89,90]. Further research with larger patient groups is needed to fully develop and refine CLAP for PJIs. However, it appears to be a promising technique in the fight against methicillin-resistant staphylococci.

## 10. Bacteriophage Therapy

### 10.1. Bacteriophage Therapy as a Way of Fighting Drug-Resistant Bacteria

Bacteriophage therapy is a novel approach to combating antibiotic-resistant bacterial species. In recent years, many studies have been conducted to further explore this treatment method. Although it is not yet widely authorized in most countries, its use is increasing, particularly for treating chronic infections caused by antibiotic-resistant bacteria. While bacteriophage therapy has not yet achieved universal acceptance or approval for routine use, there is growing global interest in its potential, especially in the context of rising antibiotic resistance. In the last few decades, due to the widespread use of antibiotics around the world, the number of multidrug-resistant bacteria infections has increased dramatically including methicillin-resistant staphylococci (MRS) infections. In order to battle this problem, bacteriophage therapy emerged as a potential way of dealing with such infections [91]. Bacteriophage is a type of virus which attacks bacterial cells. Phages occur in various environments; usually, we can find them in sewage, soil, and water habitats. There are twelve discovered phage groups that can be isolated and applied to eradicate specific bacterial species. Bacteriophages are unable to enter mammalian cells which prevents them from attacking human cells during the course of such therapy [92,93]. We can categorize phages into the following two types: lytic phages and temperate phages. When a bacteriophage infects a bacterial cell, it can follow one of two pathways. In the lytic cycle, the phage replicates within the bacterial cell, eventually causing the cell to burst (lysis) and release new phages that can infect other cells, as presented in Figure 4. In contrast, during lysogeny, the phage genome integrates with the bacterial genome, allowing the phage to replicate and be released without killing the host. Lytic phages exclusively follow the lytic cycle, while temperate phages typically undergo lysogeny but can switch to the lytic cycle if conditions change. When it comes to combating resistant bacteria, lytic phages are typically the most effective [94].

### 10.2. Bacterial Phage Defense Mechanisms and Bacteriophage Therapy Against Bacterial Biofilms

Although phages seem like a perfect solution for fighting bacteria, bacterial cells have developed several mechanisms to combat phage infections in their biofilm structures. The most common one is the modification of the biofilm surface, which prevents phage adsorption. Other mechanisms, such as CRISPR-Cas and R-M systems, degrade phage DNA genomes, preventing replication. Sie and DISARM systems block phage genome injection. Finally, Abi systems act as the last barrier, activated after phage infection, operating by initiating the suicide of the infected cell to prevent the further spread of phages [96].

Another issue especially with PJIs is the biofilm formation that has been previously mentioned. Biofilm is a multi-species microbial structure that offers protection against harsh conditions like drying and starvation. Clinically, its resistance to antibiotics is heightened due to cells with low metabolic activity, called persister cells [97].

Foreign bodies like endoprostheses are prone to biofilm formation which makes PJIs and biofilm formation a common pair. Similarly to antibiotics, phage therapy can have difficulties in fighting biofilm-related infections due to the biofilm structure and multiple defense systems. Nevertheless, most studies see them as a very successful way of dealing with such infections, although they should definitely be implemented in addition to other forms of therapy (surgery, antibiotics, etc.) to acquire the most optimal results [98].

### 10.3. Bacteriophage Therapy in Terms of MRSA and MRSE PJIs

As mentioned before, methicillin-resistant staphylococci are a very common cause of PJIs which makes these infections difficult to treat. In such situations, phage therapy might be beneficial. This therapy is usually personalized—bacteriophages must be selected based on the patient’s bacterial isolate to ensure they are effective against the patient’s bacteria. Figure 5 presents a schematic approach to implementing bacteriophage therapy in PJI cases. When dealing with PJIs, such a therapy can be implemented in several ways, as follows: intra-articularly, intravenously, orally, by means of coated-implants, or through direct local delivery, as shown in Figure 6. Typically, a combination of these delivery methods is administered. In addition, phages can be applied in singular models (only one phage type) or in cocktail forms (a mixture of several bacteriophages).

Phage therapy still requires more attention to fully understand its place in the treatment of PJIs, but its potential has already been proved in multiple animal models. In a study conducted by Kaur et al. in 2016, a group of mice underwent a surgical procedure simulating orthopedic surgery, during which different wires were inserted into their femurs. The results showed that double-coated K-wires, coated with bacteriophages and linezolid, delivered the best treatment outcomes and demonstrated bacteriophage therapy as a promising approach [100].

In another study from 2016 by Kishor et al., 22 rabbits with MRSA-induced osteomyelitis were treated with a phage cocktail. The treatment was started either 3 or 6 weeks after infection. Both groups that received therapy showed significant improvement, suggesting bacteriophage therapy as a promising treatment for MRSA osteomyelitis [101].

Additionally to studies on animals, phage therapy has been implemented successfully in some PJI cases on human patients—Table 2.

### 10.4. Bacteriophage Therapy Treatment Protocols for PJIs

As previously mentioned, bacteriophage therapy still requires more studies to be successfully implemented in the treatment of PJIs. Currently, most cases of MRSA-caused PJIs treated with bacteriophage therapy are personalized treatments with few studies on the treatment protocols. A treatment method named PhagoDAIR was introduced by Ferry et al. In France, this procedure was applied in a few complicated cases of *S. aureus* and *P. aeruginosa* PJIs. It consists of a one-shot administration of selected GMP-bacteriophages (GMP—good manufacturing practice standards) during the DAIR procedure [106]. In France, there is also a protocol funded by the French health care ministry in CRIOAc Lyon that organizes a multidisciplinary team meeting in some very complicated PJI cases to organize a personalized treatment with bacteriophage therapy included. In the PHAGEinLYON clinic program for PJI patients, phage therapy was used alongside antibiotics. The first phage injection (30–50 mL) was given during surgery or post-surgery via sonography (5 mL), followed by additional sonography-guided injections (5 mL). The frequency and timing of injections depended on the patient’s condition, and daily intravenous phages were an option for non-surgical cases [107]. Overall, bacteriophage therapy for periprosthetic joint infections (PJIs) is currently reserved for complex cases. In Europe, it works as an experimental therapy that can be applied thanks to the Declaration of Helsinki of 1964. Treatment plans are typically developed by multidisciplinary teams, combining personalized therapy with antibiotics and surgical procedures.

### 10.5. Novel Combination of DAC^®^ Hydrogel and Bacteriophage Therapy

Another possible implementation of bacteriophage therapy for periprosthetic joint infections (PJIs) is its combination with the defensive antibacterial coating (DAC^®^) hydrogel. As mentioned earlier, the DAC hydrogel is an antimicrobial coating specifically developed to protect implanted biomaterials in orthopedics and traumatology. It has demonstrated its safety and effectiveness in reducing early post-surgical infections in arthroplasties, osteosynthesis, and revision surgeries [108].

Ferry et al. [106] implemented this approach in 2020 while treating a 49-year-old male patient with a complex history of leg trauma, including a knee megaprosthesis in 2013. The patient developed a multidrug-resistant Staphylococcus epidermidis periprosthetic joint infection (PJI), which led to two fistulas with purulent discharge and prosthesis exposure, although no loosening was observed on X-ray. As salvage therapy, the patient was offered a DAIR procedure with the localized application of a lytic bacteriophage cocktail. Although bacteriophage therapy is not yet approved by the European Medicines Agency, this treatment is permitted in France under specific conditions that the patient met. Given the complexity of the procedure, a gel carrier was essential for ensuring that the phages remained at the implant surface during skin and soft tissue coverage, leading to the idea of using the DAC^®^ hydrogel.

After the DAIR procedure, clinicians applied the magistral phage preparation within the DAC^®^ gel on the megaprosthesis surface, followed by skin and soft tissue coverage. Additionally, intravenous empirical antibiotic treatment with daptomycin (850 mg daily) and tigecycline (100 mg initial dose, then 50 mg every 12 h) was initiated post-surgery. Unfortunately, five days after surgery, the patient suffered a myocardial infarction. Due to further complications, the treatment ultimately resulted in an amputation one year later; however, these complications were not considered related to the phage therapy [109].

This case led to another application of the DAC^®^ hydrogel with bacteriophage therapy for a 42-year-old male injured during the Russian war in Ukraine.

During the treatment of his complex femur injury, he developed an infection. Microbiological analysis revealed multiple drug-resistant pathogens. Despite intensive antibiotic therapy with daptomycin, cefiderocol, and fosfomycin, experimental bacteriophage therapy was suggested, as the infection had not resolved.

A final reconstruction surgery was performed, during which the phage cocktail was administered using the DAC^®^ hydrogel method, similar to the approach implemented by Ferry et al. in 2020. Bacteriophage release into the wound was monitored through drainage fluid and serum samples at 12, 24, 48, and 72 h. The patient was monitored for 12 months, during which the wound healed without complications or reinfection, and no further surgery was needed. The X-rays and CT scans confirmed successful bone healing and consolidation after 12 months, with no side effects observed from the phage administration [110].

Although the first case resulted in amputation, it highlights a new potential approach for using phage therapy, as demonstrated in the case of the 42-year-old patient from Ukraine. With further studies, this method has the potential for broader application in the future including treatment of complex MRSA-caused PJIs.

### 10.6. Bacteriophage Therapy and Its Future in Treating PJIs

In conclusion, bacteriophage therapy has significant potential for treating some of the most complex and difficult cases of prosthetic joint infections (PJIs), including those caused by MRSA. However, as the current treatment landscape for PJIs stands, this approach is not yet widely implemented. In Europe, it can be used as an experimental treatment under the Declaration of Helsinki of 1964. In Poland, this experimental treatment is available through the project led by Professor Dr. Andrzej Górski, M.D., titled “Experimental Phage Therapy for Bacterial Infections Resistant to Antibiotic Treatment, Including MRSA Infections”. This therapy can be administered in cases of infections that have not responded to antibiotic therapy, and where all standard treatment methods have been exhausted. Patients receive this treatment in several medical centers, including one in Wrocław. With further studies on the safety and utility of phage therapy, it may be applied more broadly when it comes to battling resistant infections in the future.

## 11. Materials and Methods

We collected data on the prevention and, most importantly, the treatment of prosthetic joint infections (PJIs) caused by methicillin-resistant staphylococci. Our search included keywords such as “MRSA PJI”, “MRSE PJI”, “antibiotic therapy for MRSA PJI”, “PJI MRSA treatment”, “PJI antibiotics”, “PJI bacteriophage therapy”, and “novel treatment for MRSA PJI”. We analyzed the growing threat posed by MRSA in prosthetic joint infections and explored modern approaches to prevent and combat these complications. In our review, we evaluated traditional approaches such as antibiotic therapy and innovative treatments for complex PJIs, including bacteriophage therapy. Additionally, we considered relevant case reports to enhance the practical value of our work. Wherever possible, we relied on sources published within the last 10 years.

## 12. Conclusions

In our review, we summarized current perspectives and strategies for combating prosthetic joint infections (PJIs) caused by methicillin-resistant staphylococci, with a particular emphasis on MRSA infections. We highlighted methods for preventing such complications, including the use of specific modern endoprosthesis materials and the precautionary verification of MRSA carriage in the patient’s nasopharynx.

Additionally, based on the most recent knowledge, we provided a summary of current treatment methods for MRSA PJIs. Our work emphasizes the standard treatment approach for these infections, which primarily relies on proper antibiotic therapy.

There is significant potential in the advancement of clinical trials focused on the implementation of loaded hydrogel, which has the potential to revolutionize standard treatment protocols in surgical procedures. Moreover, additional studies are required to validate the efficacy of CLAP therapy in a larger patient cohort, particularly concerning the eradication of MRSA and MRSE.

The aim of our review is to present current and innovative strategies for managing these cases, including their practical application. The information gathered is intended to assist in selecting the optimal treatment strategy for complex methicillin-resistant *Staphylococcus* spp. PJI cases. The best approach is to treat each complex PJI case individually, beginning with standard methods and transitioning to experimental treatments if the standard approach fails.

## Figures and Tables

**Figure 1 antibiotics-13-01151-f001:**
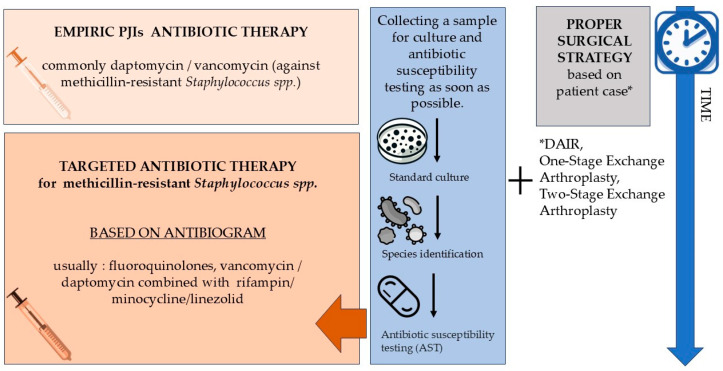
Standard treatment protocol for PJIs concerning infections caused by methicillin-resistant *Staphylococcus* species.

**Figure 2 antibiotics-13-01151-f002:**
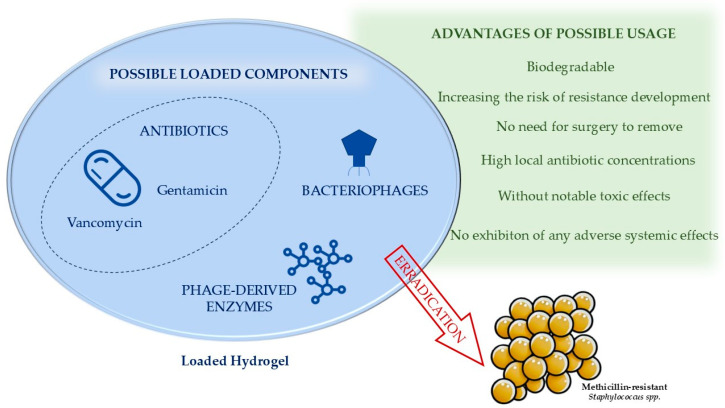
Advantages of possible usage of loaded hydrogel with example loaded components against methicillin-resistant *Staphylococcus* spp.

**Figure 3 antibiotics-13-01151-f003:**
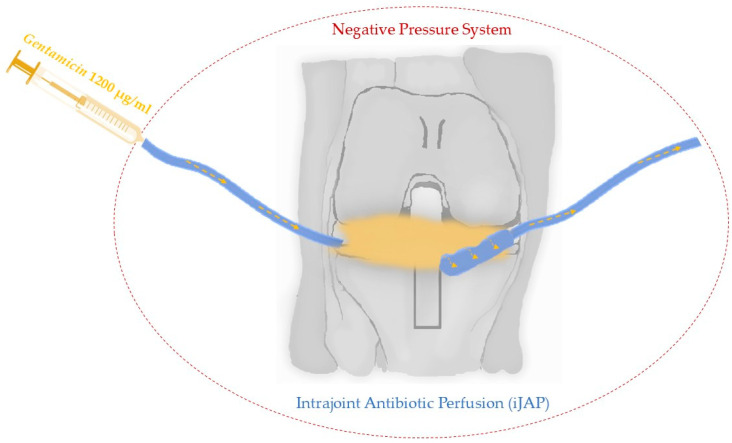
Continuous local antibiotic perfusion (CLAP)—basic structure of iJAP with typical use for MRSA antibiotic–gentamicin.

**Figure 4 antibiotics-13-01151-f004:**
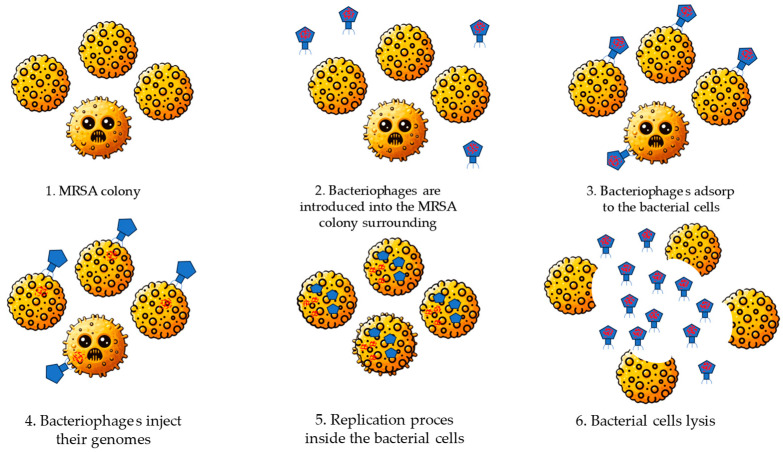
Bacteriophages lytic cycle [95].

**Figure 5 antibiotics-13-01151-f005:**
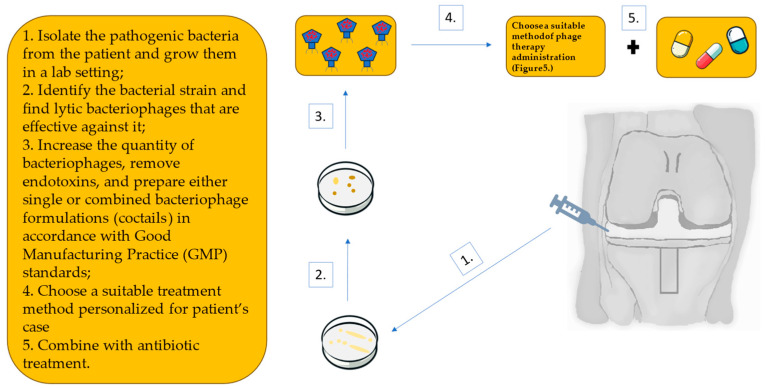
Schematic bacteriophage therapy for prosthetic joint infection (PJI) of the knee [99].

**Figure 6 antibiotics-13-01151-f006:**
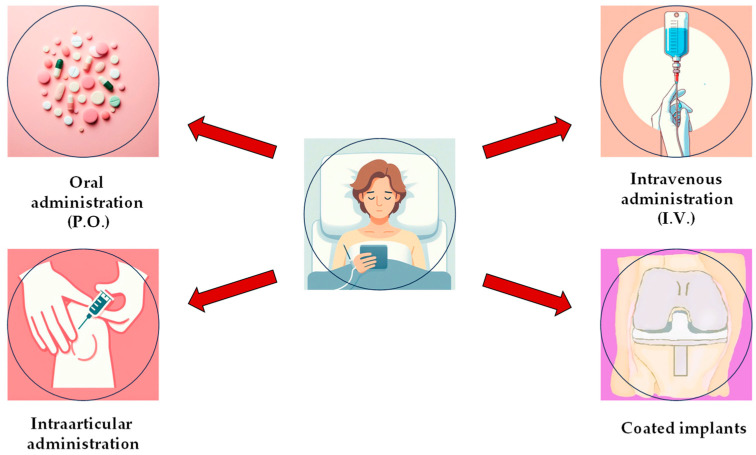
Methods of bacteriophage therapy administration.

**Table 1 antibiotics-13-01151-t001:** Penetration of antibiotics commonly used in staphylococcal infections through the biofilm extracellular matrix and into bone and joint tissues.

Bacteria	Antibiotic	Penetration Through the Biofilm Extracellular Matrix	Penetration into the Bone and Joint
*S. aures*	Amikacin	+	+
	Ciprofloxacin	+	
	Daptomycin	+	
	Oxacillin	Reduced	
	Rifampicin	+	
	Vancomycin	+/Reduced	
	Linezolid	+	
*S. epidermidis*	Amikacin	+	
	Ciprofloxacin	+	
	Daptomycin	+	
	Linezolid	+	
	Ofloxacin	+	
	Oxacillin	Reduced	
	Rifampicin	+	
	Vancomycin	Reduced	

**Table 2 antibiotics-13-01151-t002:** Bacteriophage therapy in MRSA-caused PJIs—case reports.

Patient’s History	Treatment Involving Bacteriophage Therapy	Treatment Results	Author
A 64-year-old woman with a history of right hip and knee replacements in 2018 developed a complicated MRSA infection in 2020, starting with right foot cellulitis and a dorsal abscess. Despite initial vancomycin and linezolid treatment, MRSA spread to her right knee and hip, requiring multiple surgeries, including knee prosthesis removal and hip debridement. She received IV daptomycin but could not tolerate rifampin. Despite aggressive treatments, including a two-stage revision with an antibiotic-loaded PROSTALAC and daptomycin, the MRSA infection persisted.	The patient was suggested for a personalized bacteriophage therapy (SaWIQ0488ø1 phage was chosen) for her MRSA infection, starting during surgery and continuing intravenously post-op until liver enzyme elevation required stopping the treatment. Daptomycin was also administered for 3 weeks. A second course of bacteriophage therapy was given during subsequent reconstructive hip and knee surgeries.	Patient remained free of the previously persistent MRSA infection 11 months after the first bacteriophage therapy administration.	Schoeffel et al., 2018 [102]
A patient with a femoral fracture under an endoprostheses of the hip that developed an MRSA infection—PJI.	Bacteriophage therapy using the commercial anti-*S. aureus* suspension was applied during surgery by thoroughly flooding the infection site and continued via catheter for 10 days post-operatively.	There was a full recovery within 1 year additionally to hip endoprostheses retention and osteosynthesis of the fracture.	Patey et al., 2018 [103]
A 72-year-old man with morbid obesity and hyperlipidemia had a persistent MRSA infection in his right knee prosthetic joint. He previously had MSSA septic arthritis and multiple surgeries, including a knee arthroplasty in 2012, followed by repeated infections. Despite long-term antibiotics, including vancomycin, daptomycin, and two years of oral doxycycline, the infection recurred. Due to poor healing of a femur fracture and his refusal of amputation, he sought a second opinion at the University of Maryland Medical Center.	The patient underwent prosthesis removal with placement of a static vancomycin–tobramycin spacer. Personalized bacteriophage therapy (SaGR51φ1 phage was chosen) was administered intraarticularly and intravenously, alongside a 6-week course of IV daptomycin.	After intra-articular and IV bacteriophage therapy combined with 6 weeks of daptomycin, the patient was successfully healed. Revision surgery showed no signs of infection, with negative cultures. In terms of the side effects, liver enzymes normalized soon after the IV phage administration was stopped.	Doub et al., 2020 [104]
A 70-year-old woman with hypertension, diabetes, and chronic right leg lymphedema had a persistent MRSA infection following right knee arthroplasty. Despite revision surgery and prolonged antibiotics, she developed worsening symptoms and a draining sinus, with MRSA confirmed. Previous right hip arthroplasty and femur fracture added complexity. Further revision surgery was unlikely to succeed, so knee fusion or amputation was recommended.	The patient’s MRSA isolate was treated with the bacteriophage Mallokai, administered locally during hip and knee prostheses removal surgery. Phage administration was continued by intravenous therapy, which was stopped due to elevated liver enzymes. A 6-week course of ceftaroline was also given.	A follow-up arthrocentesis showed no signs of infection. A month later, her hip and knee were reconstructed with a total femur implant, with no MRSA detected. A total of 12 months after treatment, the patient remained free of infection and was undergoing the rehabilitation process.	Doub et al., 2022 [105]

## Data Availability

Not applicable.

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
