# Peer review of "Prevention and Modern Strategies for Managing Methicillin-Resistant Staphylococcal Infections in Prosthetic Joint Infections (PJIs)"

_antibiotics, 2024, doi:10.3390/antibiotics13121151_

Round 1
Reviewer 1 Report
Comments and Suggestions for Authors
file joint

Author Response
Thank you for reviewing our work! Here are our answers to the suggestions made:
-Page 2, part 2. MRS
You could add a word on MecA and MecC resistance genes important to detect MRSA due to rapid
PCR tests. – Corrected.
L70-71, you may add a sentence/reference on CNS resistances. - Corrected.
-Page 5, part 6. Antibiotic therapy
In antibiotic therapy against MRSA and MRSE, you may add the Dalbavancin treatment and a
reference on PJI treated by this recent antibiotic. - Thank you for this valuable suggestion. We have made the necessary corrections, specifically highlighting the advantages of dalbavancin.
L256, you may detail different classes of antibiotics and the bone penetration. - We have added this information.
Table 1 is very interesting and some details could be added: what about clindamycin,
sulfamethoxazole or tetracyclines?
Penetration in bone and joint is variable between antibiotic classes, it would be important to complete
the differences in the table (ex: rifampicin and vancomycin…) - Thank you for this valuable suggestion. The table is based on studies conducted through a comparative analysis of the antibiotics listed. We believe that including information about other antibiotics from completely different publications could negatively affect the credibility of the presented content. Therefore, we decided not to expand the data in the table.
What about delafloxacin? You may add a sentence on this new molecule. - Of course. We have added a mention of this important antibiotic in the paragraph regarding MRSA, MRSE targeted antibiotics.
L283: the duration treatment is 4-6 weeks for total treatment, not only the IV part, you may modify
this sentence. - We have modified the content.
-Page 7:
L296, this part is very important, you may add a reference (combination therapy) - The reference is the work of Lora Tamayo et al.
L301, studies on cases with 1-3 months prosthesis with suspected infection is not clear, it would be
accurate to discuss about it. - Indeed, there was a lack of reference to early PJI. We have updated the paragraph.
-Page 8:
Figure 1, time of surgery is covering empiric therapy and targeted therapy. It is rare that targeted
therapy is started at time of surgery. It would be more consensual to adjust time of surgery on
empirical treatment only.
You may suppress “+another antibiotics” if you don’t detail them
In the targeted antibiotic therapy part, you may add some steps: standard culture, species
identification then AST.
If molecules are suggested, all possible antibiotic classes are to be noted, fluoroquinolones first. - We agree that the graphic was inaccurate. We have made the suggested corrections.
L346: you may add a recent reference on this part (ref 55 is published in 2011) - We have changed the reference.
-Page 10:
L433: could you precise if some relapses are observed? Are there resistant strains mentioned?
These questions are crucial on the hydrogel use. -We have expanded the topic by adding an animal study using MRSA. Since this is an innovative method, there are no data yet on its effectiveness in a statistically significant study group.
-Page 12:
CLAP is a very interesting strategy to treat PJIs. This treatment is very recent but it would be accurate
to add a point on the indications: what population, revisions surgeries only… - We agree that this was missing. We have reviewed the literature and incorporated specific references based on the current applications of CLAP in clinical practice.
Are there data on blood concentrations of antibiotics compared to local concentrations? This point
may be interesting to discuss. -Thank you for this comment. Unfortunately, there are no available standardized guidelines for such procedures, but we have added a note on the necessity of monitoring to prevent treatment toxicity, based on the experiences from the analysis by Kosugi et al.
Reviewer 2 Report
Comments and Suggestions for Authors
This manuscript reviews the management and treatment of infections caused by MRSA strains in Prosthetic Joint Infections.
The following comments are made:
1- Line 41. Put gene names in italics. Correct throughout the text.
2- Line 59. The correct word is SCCmec. Correct throughout the text.
3- Line 189. Section 5. Many decolonization programs fail because they do not take into account the colonization and persistence of S. aureus in the pharynx. Add information on the colonization, persistence and decolonization of S. aureus in the pharynx.
4- Line 250. “et al. Found”. The author is missing. Add it.
5- Figure 1. It is not cited in the text. Cite it.
6- Figures 3,4,5 and 6. They are not cited in the text. Cite them.
7- Lines 694-697. Put this sentence in the Introduction
Author Response
Thank you for reviewing our work! Here are our answers to the suggestions made:
1- Line 41. Put gene names in italics. Correct throughout the text. - corrected
2- Line 59. The correct word is SCCmec. Correct throughout the text. – corrected
3- Line 189. Section 5. Many decolonization programs fail because they do not take into account the colonization and persistence of S. aureus in the pharynx. Add information on the colonization, persistence and decolonization of S. aureus in the pharynx.
- We have added information about the importance of this aspect.
4- Line 250. “et al. Found”. The author is missing. Add it. - corrected
5- Figure 1. It is not cited in the text. Cite it. - corrected
6- Figures 3,4,5 and 6. They are not cited in the text. Cite them. – corrected
7- Lines 694-697. Put this sentence in the Introduction – The sentence has been put in the introduction.
Reviewer 3 Report
Comments and Suggestions for Authors
This review on modern prevention strategies for managing methicillin-resistant Staphylococcal infections in prosthetic joint infection is very interesting because it considers older antibiotic administration techniques and new methods for eliminating resistant Staphylococcal infections.
The classic methods of antibiotic administration are reported, as well as
the use of temporary prostheses coated with antibiotic hydrogel (particularly i
interesting technique) easy to prepare for use with different antibiotics.
Particularly, the discussion on the use of bacteriophage therapy is interesting, I believe it will have great development in the future given the shortage
in new antibiotics.
Author Response
Thank you for reviewing our work and for your kind words. We wanted to highlight technologies such as gels, CLAP, and bacteriotherapy, as we also see great potential in these therapies for the future. We hope that our work will provide valuable insights to its readers and draw attention to non-standard treatment methods, especially in the context of more challenging medical cases that do not respond to conventional treatment protocols.
Reviewer 4 Report
Comments and Suggestions for Authors
Thanks for giving me an opportunity to review. It is a good study. Here are some suggestions to make it better.
- The topic of prosthetic joint infection is well introduced and methods to deal with it are also detailed.
- Feel there is too much text. Would it be possible to summarize each treatment modality as tables. Write each study as a point and the most important finding.
- Would it be possible to comment on the consensus on the best method of treatment?
- Is there any research on treating PJI in developing countries.
Comments on the Quality of English LanguageLine 549: acquire is misspelt.
Line 375: their is misspelt.
Please review your study for grammatical errors and spelling checks.
Author Response
Thank you for reviewing our work! Here are our answers to the suggestions made:
- Feel there is too much text. Would it be possible to summarize each treatment modality as tables. Write each study as a point and the most important finding. - In the section on bacteriophage therapy, we conducted animal studies in a condensed format, highlighting only the most important aspects, as we felt there was too much text. We believe the remaining information in our review holds significant value in explaining all possible treatment methods. The tables and graphs in our figures provide a schematic summary of the treatments discussed.
- Would it be possible to comment on the consensus on the best method of treatment? - It is hard to decide on the best singular method. We believe that the best treatment approach is to consider each MRSA PJI case individually, as these infections are usually complex, and to select the most appropriate treatment method for the patient – starting with standard approaches and switching to experimental methods if the standard treatment fails. Therefore, we included the following statement in the conclusion: 'The best approach is to treat each complex PJI case individually, beginning with standard methods and transitioning to experimental treatments if the standard approach fails.
- Is there any research on treating PJI in developing countries. - We agree that this is a very important aspect of the global treatment of PJI cases; however, the main focus of this review is on the modern and most current approaches to treating MRSA prosthetic joint infections. We have explored the topic of treatment protocols and strategies in developing countries in our previous work: Mikziński P, Kraus K, Widelski J, Paluch E. Modern Microbiological Methods to Detect Biofilm Formation in Orthopedics and Suggestions for Antibiotic Therapy, with Particular Emphasis on Prosthetic Joint Infection (PJI). Microorganisms 2024, 12(6), 1198. https://doi.org/10.3390/microorganisms12061198.
Line 549: acquire is misspelt. - corrected
Line 375: their is misspelt. - corrected
Please review your study for grammatical errors and spelling checks. – corrected